# Predictors of health facility childbirth among unmarried and married youth in Uganda

**Peninah Agaba**[1]*, **Monica Magadi**[2], **Bev Orton**[2]

**1** Department of Population Studies, School of Statistics and Planning, College of Business and Management Sciences, Makerere University, Kampala, Uganda, **2** Department of Criminology and Sociology, Faculty of Arts, Cultures and Education, University of Hull, Hull, United Kingdom

* peninah.agaba@mak.ac.ug, agabapeninah@gmail.com

**Data Availability Statement:** The data underlying the results presented in the study are available from the measure dhs website on https://dhsprogram.com/data/available-datasets.cfm

## Abstract

### Background

Uganda has a high maternal mortality rate combined with poor use of health facilities at childbirth among youth. Improved use of maternal health services by the youth would help reduce maternal deaths in the country. Predictors of use of health facilities at childbirth among unmarried compared to married youth aged 15–24 years in Uganda between 2006 and 2016 are examined.

### Methodology

Binary logistic regression was conducted on the pooled data of the 2006, 2011 and 2016 Uganda Demographic and Health Surveys among youth who had given birth within five years before each survey. This analysis was among a sample of 764 unmarried, compared to 5,176 married youth aged 15–24 years.

### Results

Overall, unmarried youth were more likely to have a childbirth within the health facilities (79.3%) compared to married youth (67.6%). Higher odds of use of health facilities at childbirth were observed among youth with at least secondary education (OR = 2.915, 95%CI = 1.747–4.865 for unmarried *vs* OR = 1.633, 95%CI = 1.348–1.979 for married) and frequent antenatal care of at least four visits (OR = 1.758, 95%CI = 1.153–2.681 for unmarried *vs* OR = 1.792, 95%CI = 1.573–2.042 for married). Results further showed that youth with parity two or more, those that resided in rural areas and those who were engaged in agriculture had reduced odds of the use of health facilities at childbirth. In addition, among married youth, the odds of using health facilities at childbirth were higher among those with at least middle wealth index, and those with frequent access to the newspapers (OR = 1.699, 95% CI = 1.162–2.486), radio (OR = 1.290, 95%CI = 1.091–1.525) and television (OR = 1.568, 95%CI = 1.149–2.138) compared to those with no access to each of the media, yet these were not significant among unmarried youth.

**Funding:** The authors received no specific funding for this work.

**Competing interests:** The authors have declared that no competing interests exist.

## Conclusion and recommendations

Frequent use of antenatal care and higher education attainment were associated with increased chances of use of health facilities while higher parity, rural residence and being employed in the agriculture sector were negatively associated with use of health facilities at childbirth among both unmarried and married youth. To enhance use of health facilities among youth, there is a need to encourage frequent antenatal care use, especially for higher parity births and for rural residents, and design policies that will improve access to mass media, youth's education level and their economic status.

## Background

Despite the global progress in improving maternal mortality, women in sub-Saharan Africa (SSA) are 14 times more likely to die during the maternity period compared to women in developed regions [1, 2]. In Uganda, about 16 women die each day due to maternal mortality [3]. Most maternal deaths occur during labour, at childbirth or within two days postpartum, [1, 4–8], this highlights that most deaths would be averted with prompt and adequate diagnosis, and care during and after the childbirth process [2, 9, 10]. The use of health facilities has been associated with reduced maternal mortality [6, 9, 11–13]. Childbirth in a health facility is an opportunity to aid a mother during childbirth, supervise her immediately after childbirth and identify, manage and/or prevent complications [14].

Researchers have suggested that for countries to achieve SDG3, target 3.1, the use of maternal health services should be universal [15] or the use of health facilities at birth should be at least 81 percent together with a 91 percent use of at least one antenatal care visit, 78 percent of four antenatal care visits, and 87 percent of skilled birth attendance [7]. However, despite the correlation of the use of health facilities with maternal survival, its use remained low in Uganda (41% in 2006 & 57% in 2011 [16, 17]. However, a greater increase was observed in the five years before the 2016 UDHS from 57 percent in 2011 to 73 percent in 2016 [3, 17]. Coupled with poor use of antenatal care and postnatal care, Uganda is unlikely to achieve SDG 3.1 [16, 17].

Although younger women aged 15–24 years generally have higher odds of using health facilities at childbirth than older women (29), there are notable variations among young women. For instance, older adolescents and youth tend to be more likely to use health facilities at childbirth than very young youth below 17 years [18–20], which could be attributed to differences in knowledge and support to access health facilities at childbirth [21]. More importantly, use of health facilities at childbirth was found to be higher among married than unmarried youth due to various societal and health system factors including husband support, shame and discrimination among unmarried youth [22–24].Studies have also documented that working away from home [25] and higher parity [25–31] are associated with reduced chances of use of health facilities at childbirth among youth. In addition, higher education level [19, 23, 25–29, 31–33], higher wealth status [18, 20, 23, 25, 26, 28, 29, 31, 32], urban residence [18–20, 23, 25, 27, 31–33], access to mass media [20, 29], and early, frequent and quality antenatal care [23, 25–27, 29, 30, 33] have been associated with increased chances of giving birth from health facilities by youth. Furthermore, good community perspectives about health provider skills were associated with improved use of health facilities at childbirth among married adolescents in India [34]. However, in contrast, a study in Nigeria found no significant

difference in health facility use at childbirth by wealth index among adolescents [29]. Furthermore, a study in Malawi found that higher parity was associated with increased odds of use of health facilities at childbirth [35].

Studies have documented the factors for use of health facilities among youth [19, 23, 25–29, 33, 36–38] but no population-based studies were found to have been carried out comparing unmarried and married youth. The situation is dire for pregnant unmarried youth who have their own unique social, economic, psychological, health and obstetrical needs; and their experiences during the maternity period especially, abuse and stigmatisation puts them in a vulnerable position [36, 39–41]. This study used a nationally representative survey to find out the variations in factors associated with the use of health facilities at childbirth among unmarried compared to married youth aged 15–24 years in Uganda. This is aimed at providing information to guide health providers and policy makers to overcome barriers and improve the use of health facilities at childbirth, which will subsequently reduce maternal mortality among the youth. The main objective of this study is to examine the factors associated with the use of health facility at childbirth among unmarried compared to married youth in Uganda.

## Methods

### Source of data

Secondary analysis of the pooled 2006, 2011 and 2016 Uganda Demographic and Health Survey (UDHS) data was conducted. The UDHS data was retrieved with permission from MEASURE DHS (dhsprogram.com). These surveys were population-based household surveys that used two stage sampling where clusters (Enumeration Areas) and households from each cluster were randomly selected. Detailed sampling procedures are available in respective UDHS reports [3, 16, 17]. The UDHSs collect data on socio-economic and demographic characteristics of women, and their use and timing of maternal and child health services, and nutrition among other factors. Young women aged 15–24 years who had had a birth five years before each survey were selected for this analysis. These criteria resulted into a sample of 5,940 cases of which 764 were unmarried and 5,176 were married youth.

### Measure of the outcome variable

The dependent variable is the use of a health facility at childbirth which was coded as Yes = 1 if the youth gave birth in the health facility and No = 0 if she gave birth elsewhere including at home, on the road, or at the traditional birth attendants' place.

### Measures of predictor variables

Based on previous empirical literature and their availability in the UDHS data, independent factors that were included in the analysis are presented in the Table 1 below.

**Statistical analysis.** Data were analysed at the univariate, bivariate, and multivariate levels using SPSS 24. Descriptive statistics of the background characteristics of the respondents were presented at univariate level. At bivariate level, Pearson's chi-squared (χ2) tests were used to examine significant differences between health facility childbirth and the explanatory variables, and the trends in health facility childbirth over the years. Binary logistic regression models were fitted to find the factors associated with health facility childbirth among unmarried compared to married youth. The multivariate model included explanatory variables whose p-values were less than 0.05 during the chi-square tests, except for age group and survey year. Results are presented in the form of odds ratios (OR) with 95% confidence intervals of ORs.

**Table 1. Description of predictor variables.**

| Name of predictor variable | Measure |
|---|---|
| Age | Coded as 15–19 and 20–24 years |
| Parity | A dichotomous variable coded as 1 if the respondent had one child and 2 if the respondent had two or more children |
| Pregnancy desire | A dichotomous variable coded as 1 if the respondent wanted to get pregnant then and 2 if she did not want to get pregnant or wanted to get pregnant later |
| Sex of household head | Coded as 1 = if respondent was living in a male-headed household at the time of the survey; 0 = otherwise. |
| Number of ANC visits | Coded as 1 if the respondent had no or less than four ANC visits and 2 if the respondent had at least four ANC visits |
| Education level | Dummy variables for highest educational attainment classified into two categories: no education or primary education and secondary and above |
| Religion | Dummy variables for religious affiliation, re-coded into four categories: 1. Catholics, 2. Protestants, 3. Muslims 4. Other religions |
| Wealth index | Dummy variables for DHS household wealth index that is developed from household assets and constructed by principle component analysis. The PCA scores are classified as poorest, poorer, middle, richer, and richest wealth quintiles |
| Occupation | Dummy variables for occupation recoded as not working, those employed in the agriculture sector, professionals, and labourers |
| Place of residence | Coded as 1 if the respondent was residing in rural areas and 2 if the respondent was residing in urban areas |
| Region | Dummy variables for region coded into four categories: central, east, north and west |
| Access to newspapers | Dummy variables for access to newspapers categorised into three categories as no access, less frequent access or some access and more frequent access or daily access |
| Access to radio | Dummy variables for access to radio categorised into three categories: no access, less frequent access or some access and more frequent access or daily access |
| Access to television | Dummy variables for access to television categorised into three categories as no access, less frequent access or some access and more frequent access or daily access |

The data was weighted using the individual/women weights (v005) to account for the complex survey design that is applied in DHS data collection and non-response. This data was tested for multi-collinearity of the variables using tolerance and the variance inflation factor (VIF). Multicollinearity test results showed that none of the variables in the model had a tolerance threshold of less than 0.10 or a VIF of 10; actually, all VIF values were below 2. The goodness of fit was tested using the Hosmer and Lemeshow test and the models were good fit as the Hosmer-Lemeshow chi-squared p-values were greater than 5% for both unmarried and married youth.

## Research ethics statement

No approval for using UDHS data was required since UDHS is a secondary data source and available in the public domain for use with no identifiable information about participants. However, to access the data, we sought permission from MEASURE DHS. Strict ethical requirements were observed during UDHS data collection. Informed consent was sought from all participants before each interview in the UDHS. ICF institutional review board and a local institutional review board in Uganda approved the UDHS questionnaires. ICF IRB ensured that the survey complied with the U.S. Department of Health and Human Services regulations for the protection of human subjects, while the Uganda's IRB ensured that the survey complied with local laws and norms. Privacy, anonymity, and confidentiality were ensured during the interviews, data storage and analysis. Details about UDHS data and ethical standards are

available at: https://dhsprogram.com/What-We-Do/Protecting-the-Privacy-of-DHS-Survey-Respondents.cfm

## Results

### Descriptive characteristics of the respondents

Table 2 shows that more than half of both the unmarried and married respondents lived in rural areas (69% vs 86%), were aged 20–24 years (56% vs 76.5%), and had no or had attained primary education (54% vs 84%). More than half had had four or more ANC visits among

**Table 2. Distribution of unmarried and married youth by background factors.**

| Variable | Frequency (percentage) | Number, Health facility childbirth | Percentage health facility childbirth | Frequency (percentage) | Number, Health facility childbirth | Percentage health facility childbirth |
|---|---|---|---|---|---|---|
| | Unmarried youth | | | Married youth | | |
| Survey year | | | P = 0.000 | | | P = 0.000 |
| 2006 | 171(22.7) | 112 | 65.5 | 1,313(25.2) | 638 | 48.6 |
| 2011 | 169(22.4) | 118 | 69.8 | 1,227(23.5) | 791 | 64.5 |
| 2016 | 414(54.9) | 368 | 88.9 | 2,675(51.3) | 2,096 | 78.4 |
| Age | | | P = 0.115 | | | P = 0.015* |
| 15–19 | 334(43.7) | 253 | 77.1 | 1053(20.3) | 737 | 70.5 |
| 20–24 | 430(56.3) | 343 | 81.0 | 4123(79.7) | 2789 | 66.9 |
| Pregnancy wanted | | | P = 0.519 | | | P = 0.274 |
| Then | 208(27.2) | 156 | 80.8 | 3085(59.6) | 2102 | 68.2 |
| Later or not anymore | 556(72.8) | 442 | 78.6 | 2091(40.4) | 1424 | 66.8 |
| Birth order/Parity | | | P = .000* | | | P = 0.000* |
| One | 654(85.6) | 534 | 82.8 | 2038(39.4) | 1579 | 76.8 |
| Two or more | 110(14.4) | 65 | 58.6 | 3138(60.6) | 1947 | 61.6 |
| ANC numbers | | | P = 0.000* | | | P = 0.000* |
| None or 1–3 visits | 352(46.6) | 256 | 72.7 | 2323(44.5) | 1341 | 57.5 |
| 4+ visits | 403(53.4) | 342 | 84.9 | 2892(55.5) | 2185 | 75.6 |
| Sex of household head | | | P = 0.085 | | | P = 0.064 |
| Male | 358(47.4) | 276 | 77.1 | 4448(85.3) | 2960 | 66.6 |
| Female | 397(52.6) | 322 | 81.3 | 768(14.7) | 565 | 73.7 |
| Education level | | | P = 0.000* | | | P = 0.000* |
| No education or Primary Education | 413(54.1) | 275 | 69.6 | 3896(75.3) | 2336 | 61.0 |
| Secondary | 351(45.9) | 324 | 90.0 | 1280(24.7) | 1190 | 85.7 |
| Religion | | | p = 0.041* | | | P = 0.000* |
| Catholic | 291(38.1) | 218 | 78.4 | 1775(34.3) | 1205 | 67.8 |
| Protestant | 261(34.2) | 197 | 74.9 | 1903(36.8) | 1190 | 63.1 |
| Muslims | 110(14.4) | 98 | 86.0 | 773(14.9) | 624 | 77.4 |
| Others | 102(13.4) | 85 | 85.0 | 725(14.0) | 507 | 68.1 |
| Type of Residence | | | P = 0.000* | | | P = 0.000* |
| Urban | 234(30.6) | 193 | 91.9 | 941(18.2) | 853 | 89.9 |
| Rural | 530(69.4) | 405 | 74.3 | 4235(81.8) | 2673 | 62.7 |
| Region | | | p = 0.035* | | | P = 0.000* |
| Central | 259(33.9) | 235 | 83.3 | 1177(22.7) | 1087 | 79.9 |
| Eastern | 186(24.3) | 149 | 81.4 | 1507(29.1) | 986 | 63.6 |
| Northern | 147(19.2) | 81 | 73.6 | 1362(26.3) | 673 | 64.2 |

*(Continued)*

**Table 2.** (Continued)

| Variable | Frequency (percentage) | Number, Health facility childbirth | Percentage health facility childbirth | Frequency (percentage) | Number, Health facility childbirth | Percentage health facility childbirth |
|---|---|---|---|---|---|---|
| Western | 164(22.5) | 133 | 73.9 | 1130(21.8) | 779 | 62.2 |
| **Wealth index** | | | *p = 0.001** | | | *P = 0.000** |
| Poorest | 116(15.2) | 66 | 68.0 | 1339(25.9) | 676 | 56.1 |
| Poorer | 104(13.6) | 76 | 77.6 | 1255(24.2) | 750 | 59.9 |
| Middle | 132(17.3) | 102 | 76.7 | 903(17.4) | 610 | 64.8 |
| Richer | 164(21.5) | 137 | 77.8 | 771(14.9) | 600 | 71.2 |
| Richest | 248(32.5) | 216 | 87.1 | 908(17.5) | 890 | 91.3 |
| **Woman's Occupation** | | | *p = 0.000** | | | *P = 0.000** |
| Not working | 244(31.9) | 212 | 86.5 | 1119(21.6) | 891 | 77.7 |
| Agriculture | 263(34.4) | 177 | 67.0 | 2712(52.4) | 1606 | 58.0 |
| Labourers | 69(9.0) | 56 | 87.5 | 464(9.0) | 311 | 75.1 |
| Professionals | 188(24.6) | 153 | 84.1 | 880(17.0) | 718 | 81.2 |
| **Frequency of reading newspapers** | | | *P = 0.006** | | | *P = 0.000** |
| Not at all | 531(69.5) | 396 | 76.2 | 4274(82.6) | 2739 | 64.1 |
| Less frequent | 116(15.2) | 100 | 87.0 | 574(11.1) | 494 | 80.9 |
| More frequent | 117(15.3) | 102 | 85.7 | 328(6.3) | 293 | 88.0 |
| **Frequency of listening to the radio** | | | *P = 0.642* | | | *P = 0.000** |
| Not at all | 176(23.0) | 130 | 76.9 | 1143(22.1) | 665 | 60.1 |
| Less frequent | 106(13.9) | 78 | 78.8 | 655(12.7) | 398 | 66.9 |
| More frequent | 482(63.1) | 391 | 80.3 | 3378(65.3) | 2462 | 70.1 |
| **Frequency of watching TV** | | | *P = 0.001** | | | *P = 0.000** |
| Not at all | 484(63.4) | 352 | 74.9 | 3994(77.2) | 2514 | 62.4 |
| Less frequent | 97(12.7) | 83 | 84.7 | 493(9.5) | 346 | 75.4 |
| More frequent | 183(24.0) | 163 | 87.6 | 689(13.3) | 666 | 91.1 |
| **Total** | **N = 754(100)** | **598** | **79.3%** | **N = 5176(100)** | **3,526** | **67.6%** |

*Statistical significance at 5% level p<0.05

Not at all- No access to any, Less frequent- once or less than once; More frequent- Almost daily access

both unmarried (53.4%) and married (55.5%) youth. The main source of information for both samples was the radio (63% unmarried compared to 65% among married youth) However, a larger proportion of unmarried youth wanted the pregnancy later (73%) and had only one birth (86%); compared to the married who wanted the pregnancy then (60%) and had had two or more births (61%).

Table 2 shows an increase in the levels of the use of health facilities at childbirth between the survey years among youth. The proportions were higher among the unmarried than the married across all the survey years. Use of health facility at childbirth was high among unmarried youth in 2016 (89%) compared to unmarried youth in 2006 (60%). The proportion of health facility childbirth was high among married youth in 2016 (78%) compared to married youth in 2006 (49%). A greater increment of 30% was among the married compared to 24% among unmarried youth. This increasing trend was significant at p = 0.000 among unmarried and married youth.

Table 2 further shows the cross tabulations between health facility use at childbirth and each independent variable among unmarried and married youth.

**Unmarried youth.** There was a significant relationship between parity (p = 0.000), frequent ANC use (p = 0.000), education level (p = 0.000), religion (p = 0.041), place of residence (p = 0.000), region of residence (p = 0.035), occupation (p = 0.000), wealth index (p = 0.001), access to newspapers (p = 0.006), and television (p = 0.001) with health facility childbirth among unmarried youth.

**Married youth.** Age group (p = 0.015), parity (p = 0.001), frequent ANC use (p = 0.001), education level (p = 0.001), religion (p = 0.001), place of residence (p = 0.001), region of residence (p = 0.001), education level (p = 0.001), occupation (p = 0.001), wealth index (p = 0.001), and access to media ((newspapers (p = 0.0001), radio (p = 0.001), television (p = 0.001)) had a significant relationship with health facility childbirth among married youth.

On the other hand, this analysis found that there was no significant difference in the use of health facilities at childbirth by pregnancy desire and sex of household head for both unmarried and married youth.

**Predictors of health facility use at childbirth among unmarried compared to married youth in Uganda between 2006–2016.** The education level of participants, parity, occupation, place of residence, and antenatal care use were found to be significantly associated with use of health facilities at childbirth among both unmarried and married youth. In addition, wealth index and access to the radio, newspapers and television were significantly associated with use of health facilities at childbirth among married youth (Table 3).

Unmarried youth with at least secondary level of education were about three times more likely to deliver in a health facility compared to counterparts with no or primary level education (OR = 3.45, 95%CI = 2.85–4.16) whereas married youth with secondary level education were 63% more likely to deliver in a health facility (OR = 1.633, 95%CI = 1.348–1.979).

Higher parity was negatively associated with health facility use among both unmarried and married youth. Youth with at least two children were less likely to give birth in health facilities compared to those who were pregnant for the first time (OR = 0.284, 95%CI = 0.165–0.490 among unmarried and OR = 0.593,95%CI = 0.509–0.691 among married youth). Unmarried and married youth who were employed in the agriculture sector had reduced odds of using health facilities at childbirth compared to youth who were not employed (OR = 0.541, 95% CI = 0.314–0.932 vs OR = 0.788, 95%CI = 0.656–0.946 among unmarried and married youth respectively).

Rural residence was negatively associated with use of health facilities among both unmarried and married youth (OR = 0.416, 95%CI = 0.217–0.799 vs OR = 0.550, 95%CI = 0.423–0.7155 among unmarried and married youth respectively). The number of ANC visits and education level were significantly associated with higher odds of delivering in health facilities among both unmarried and married youth. Unmarried and married youth who had four or more ANC visits had higher odds of using health facilities at childbirth compared to those who had no ANC or had less than four ANC visits (OR = 1.758,95%CI = 1.153–2.681 among unmarried & OR = 1.792,95%CI = 1.573–2.042 among married youth).

Married youth residing in households with at least middle wealth index were more likely to give birth from health facilities compared to married youth living in poorest households (OR = 1.362, 95%CI = 1.106–1.678, OR = 1.444, 95%CI = 1.146–1.820 and OR = 3.21, 95% CI = 2.276–4.529 for middle, richer and richest wealth index respectively). Also, married youth with more frequent access to the newspapers (OR = 1.699, 95%CI = 1.162–2.486), radio (OR = 1.290, 95%CI = 1.091–1.525) and television (OR = 1.568, 95%CI = 1.149–2.138) had higher odds of using health facilities at childbirth compared to those with no access to the newspapers, radio, and television respectively.

**Table 3.** Multivariable analysis of factors associated with health facility use at childbirth among youth in Uganda.

| | Unmarried youth | | Married youth | |
|---|---|---|---|---|
| Variable | Number, Health facility childbirth (Percentage health facility childbirth) | Adjusted Odds ratio with 95% Confidence Interval | Number, Health facility childbirth (Percentage health facility childbirth) | Adjusted Odds ratios with 95% Confidence Interval |
| **Year of survey** | | | | |
| **2006** | 112(65.5) | 1 | 638(48.6) | 1 |
| 2011 | 118(69.8) | 1.204 (0.705–2.054) | 791(64.5) | 1.826 (1.523–2.190)*** |
| 2016 | 368(88.9) | 4.589 (2.678–7.864)*** | 2,096(78.4) | 3.537 (2.998–4.173)*** |
| **Age** | | | | |
| **15–19** | 253(77.1) | 1 | 737(70.5) | 1 |
| 20–24 | 343(81.0) | 1.321 (0.841–2.076) | 2789(66.9) | 0.926 (0.773–1.108) |
| **Birth order/Parity** | | | | |
| **One** | 534(82.8) | 1 | 1579(76.8) | 1 |
| Two or more | 65(58.6) | 0.284 (0.165–0.490)*** | 1947(61.6) | 0.593 (0.509–0.691)*** |
| **ANC use** | | | | |
| **No or 1–3 visits** | 256(72.7) | 1 | 1341(57.5) | 1 |
| 4+ ANC visits | 342(84.9) | 1.758 (1.153–2.681)** | 2185(75.6) | 1.792 (1.573–2.042)*** |
| **Woman's Education level** | | | | |
| **No education or Primary education** | 275(69.6) | 1 | 2336(61.0) | 1 |
| Secondary+ | 324(90.0) | 2.915 (1.747–4.865)*** | 1190(85.7) | 1.633 (1.348–1.979)*** |
| **Religion** | | | | |
| **Catholics** | 218(78.4) | 1 | 1205(67.8) | 1 |
| Protestant | 197(74.9) | 0.835 (0.521–1.338) | 1190(63.1) | 0.902 (0.774–1.052) |
| Muslim | 98(86.0) | 1.464 (0.717–2.989) | 624(77.4) | 1.201 (0.966–1.495) |
| Others | 85(85.0) | 1.356 (0.675–2.723) | 507(68.1) | 0.937 (0.764–1.150) |
| **Wealth index (RC =)** | | | | |
| **Poorest** | 66(68.0) | 1 | 676(56.1) | 1 |
| Poorer | 76(77.6) | 1.101 (0.516–2.349) | 750(59.9) | 1.117 (0.933–1.337) |
| Middle | 102(76.7) | 0.889 (0.418–1.889) | 610(64.8) | 1.362 (1.106–1.678)** |
| Richer | 137(77.8) | 0.916 (0.442–1.899) | 600(71.2) | 1.444 (1.146–1.820)** |
| Richest | 216(87.1) | 1.260 (0.534–2.973) | 890(91.3) | 3.211 (2.276–4.529)*** |
| **Woman's Occupation** | | | | |
| **Not working** | 212(86.5) | 1 | 891(77.7) | 1 |
| Agriculture | 177(67.0) | 0.541 (0.314–0.932)** | 1606(58.0) | 0.788 (0.656–0.946)** |
| Labourers | 56(87.5) | 00.777 (0.296–2.041) | 311(75.1) | 0.883 (0.658–1.186) |
| Professionals | 153(84.1) | 0.536 (0.286–1.005) | 718(81.2) | 0.949 (0.743–1.212) |
| **Place of residence** | | | | |
| **Urban** | 193(91.9) | 1 | 853(89.9) | 1 |
| Rural | 405(74.3) | 0.416 (0.217–0.799)*** | 2673(62.7) | 0.550 (0.423–0.715)*** |
| **Frequency of reading newspapers** | | | | |
| **Not at all** | 396(76.2) | 1 | 2739(64.1) | 1 |
| Less frequent | 100(87.0) | 1.691 (0.857–3.335) | 494(80.9) | 1.165 (0.911–1.489) |
| More frequent | 102(85.7) | 1.117 (0.556–2.244) | 293(88.0) | 1.699 (1.162–2.486)** |
| **Frequency of listening to radio** | | | | |
| **Not at all** | 130(76.9) | 1 | 665(60.1) | 1 |
| Less frequent | 78(78.8) | 0.884 (0.427–1.829) | 398(66.9) | 1.102 (0.870–1.397) |

*(Continued)*

**Table 3.** (Continued)

| Variable | Unmarried youth | | Married youth | |
|---|---|---|---|---|
| | Number, Health facility childbirth (Percentage health facility childbirth) | Adjusted Odds ratio with 95% Confidence Interval | Number, Health facility childbirth (Percentage health facility childbirth) | Adjusted Odds ratios with 95% Confidence Interval |
| More frequent | 391(80.3) | 0.953 (0.558–1.630) | 2462(70.1) | 1.290 (1.091–1.525)** |
| **Frequency of watching TV** | | | | |
| **Not at all** | 352(74.9) | 1 | 2514(62.4) | 1 |
| Less frequent | 83(84.7) | 1.012 (0.508–2.018) | 346(75.4) | 1.004 (0.780–1.293) |
| More frequent | 163(87.6) | 0.987 (0.518–1.880) | 666(91.1) | 1.568 (1.149–2.138)** |
| **Observations** | **598(79.3)** | **764** | **3,526(67.6)** | **5,176** |

*p<0.05

**p<0.01

***p<0.001

1 = Reference category

## Discussion

The main aim of this study was to find out the predictors of the use of health facilities at childbirth among unmarried compared to married youth, aged 15–24 years in Uganda between 2006 and 2016. This study observed the marital status variations in the use of health facilities at childbirth; unmarried youth were more likely to have their childbirth in the health facilities than married youth. Use of health facilities at childbirth was associated with education level, parity, place of residence, occupation, and frequency of ANC use among both unmarried and married youth. In addition, it was associated with wealth index, access to newspapers, radio and television among married youth.

Unmarried youth were observed to use health facilities more than the married youth in Uganda between 2006 and 2016. This could be due to the confounding effect of higher parity as most married youth were pregnant for the second time or more, and higher parity has been associated with lower use of safe childbirth in prior studies among youth [20, 25, 26, 29–31]. A separate regression analysis combining the two samples and considering marital status as one of the factors (results not presented) shows that the effect of marital status ceases to be significant when these confounding factors are controlled for, suggesting that the differences in use of health facilities at childbirth among unmarried and married youth is accounted for by the cofounding factors. Parity was significantly associated with the use of health facilities at childbirth in this study. Youth who have had two or more children were less likely to use health facilities at childbirth compared to those who were having their first birth in the current study. This has been found in previous studies among youth [25–28, 30–32]. Studies have found that women with higher parity who have had no history of pregnancy complications [42], have had better pregnancy outcomes with no infant mortality [43], and believe that they can give birth on their own without the need to go to the health facilities [44] may see no reason for using health facilities for childbirth. A separate analysis suggests that the apparent higher use of health facilities during childbirth among unmarried than married youth is largely explained by a higher proportion of higher parity births among married women.

Our findings show that youth with higher education level was associated with higher odds of giving birth in health facilities compared to those with no or primary education level. This is in line with what has been found in prior studies among youth [20, 23, 26, 28, 29, 31, 33, 45]. This could be partially linked to higher levels of knowledge on the benefits of seeking

healthcare and women empowerment which enhances their decisions concerning their health care [25, 28]. The result suggest that the effect of education is stronger for unmarried than married youth, suggesting that women's empowerment and decision making regarding their healthcare is more applicable to unmarried youth than married counterparts for whom partners and/or in-laws are likely to influence healthcare decision making [46].

This study found that rural areas were associated with lower chances of the use of health facilities at childbirth among both unmarried and married youth. This finding is consistent with previous studies that observed lower odds of the use of health facilities at childbirth in rural areas than urban areas [18, 19, 31–33]. The observed rural-urban differences could be explained by the long distances, the times, and transport means required to reach the health centres during untimed pre-birth labour pains, which are more unfavourable for rural youth than urban youth. There are also poor staffing levels of health facilities in rural areas especially with midwives [10, 47, 48].

Our results showed that the use of health facilities at childbirth was low for both unmarried and married youth employed in the agriculture sector. The negative influence of agriculture could be because it is labour intensive, and this competes with the time to access health facilities for childbirth compared to non-working youth [49]. In addition, most of the agriculture work that these young do is for their own subsistence, and only what is the excess is sold, thus less or no income is obtained from this agriculture compared to other work. Consequently, they have less money to meet the requirements for the use of health facilities at childbirth.

Economic disparities in the use of health facilities at childbirth were also identified in this study among married youth. Married youth in middle, richer and richest wealth quintiles had higher chances of the use of health facilities at childbirth, but not among unmarried youth. Several studies have shown that economic power increases the odds of using health facilities or safe childbirth among youth [18, 20, 25, 26, 28, 31, 32]. This could be explained by the affordability of the services and other indirect costs. Households with relatively high incomes can assign higher proportions of their income to health care, compared to poor households whose priority is meeting household basic needs [31]. The rich also have access to mass media, thus have better knowledge of the benefit of using health facilities at childbirth. However, it is interesting to note that household wealth is not a significant predictor of health facility childbirth for unmarried youth. This may suggest that for unmarried adolescents, social or service factors are likely to be more important than economic power in determining use of health services [24].

In the current study, access to the radio, newspapers, and television was associated with higher odds of the use of health facilities at childbirth among married youth, yet not among unmarried youth. The association between access to mass media and the use of health facilities has been soundly established in previous studies that observed that youth, including adolescents who had access to media, had increased chances of using health facilities at childbirth [20, 26–28]. Access to media increases knowledge of the benefits of the use of health facilities at childbirth, and the dangers of birth outside health facilities thus, increased chances of the use of health facilities at childbirth. Unmarried youth might encounter other barriers even when they are well informed about the benefits of use of health facilities at childbirth. Media, especially the radio, is a good source of maternal health information for Uganda, as most households have access to the radio (55.2%), compared to 7.2 percent and 2.1 percent that have access to television and print media respectively [50].

This study had some limitations that need to be acknowledged. Uganda Demographic and Health surveys (UDHS) collect data from women for births in the last five years before the date of the survey, which may lead to inaccuracies due to memory lapse. In addition, the cross-sectional nature of data from UDHS did not allow us to infer causal relationship between

health facility use at childbirth and socio-economic factors. This is because use of health facilities at childbirth is asked for a birth in the last five years, while socio-economic factors are as of the time of the interview. However, the UDHS remains one of the most robust nationally representative data sets in understanding health facility use. Thus, this study increases knowledge about the factors associated with use of health facilities at childbirth among unmarried and married youth in Uganda.

## Conclusion and recommendations

Use of health facilities at childbirth was positively related to higher education level and frequent antenatal care use. It was negatively associated with higher parity, rural residence and being employed in the agriculture sector among both unmarried and married youth. In addition, it was positively associated with higher wealth index and more frequent access to media among married youth.

To enhance the use of health facilities among youth in Uganda, there is need to encourage frequent ANC attendance, improve education, household socio-economic status and access to media. Efforts should also aim to remove barriers to the use of health facilities at childbirth for youth with higher parity, those employed in the agriculture sector, and those that reside in rural areas.

## Acknowledgments

Analysis presented in this paper was undertaken during PA's PhD research. Peninah Agaba is a Commonwealth Scholar, funded by the UK government. The authors are grateful to Makerere University, Department of Population Studies and University of Hull, Faculty of Arts and Culture Education for creating an enabling environment that made it possible to carry on this study. We thank the DHS program for granting us the permission to use the UDHS data.

## Author Contributions

**Conceptualization:** Peninah Agaba, Monica Magadi, Bev Orton.

**Formal analysis:** Peninah Agaba.

**Methodology:** Monica Magadi.

**Validation:** Monica Magadi.

**Writing – original draft:** Peninah Agaba.

**Writing – review & editing:** Peninah Agaba, Monica Magadi, Bev Orton.

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
