## [Decision Letter · Decision Letter 0]

16 Mar 2021

PONE-D-20-31194

FACTORS ASSOCIATED WITH HEALTH FACILITY DELIVERY AMONG UNMARRIED AND MARRIED YOUTH IN UGANDA

PLOS ONE

Dear Dr. Agaba,

Thank you for submitting your manuscript to PLOS ONE. After careful consideration, we feel that it has merit but does not fully meet PLOS ONE’s publication criteria as it currently stands. Therefore, we invite you to submit a revised version of the manuscript that addresses the points raised during the review process.

The manuscript has been evaluated by two reviewers, and their comments are available below. You will see the reviewers have commented on the importance of your manuscript to policy makers. However, the reviewers have also raised critical concerns and the manuscript will need significant revision before it can be considered for publication – you should anticipate that the reviewers will be re-invited to assess the revised manuscript, so please ensure that your revision is thorough. I have outlined some of the key concerns noted by the reviewers below, but you should respond to all concerns mentioned by the reviewers in your response-to-reviewers document. 

The key concerns noted by the reviewers relate to the reporting of the methods and the results. Specifically, the reviewers have requested the addition of n values in the results section. Reviewer 2 has requested clarity regarding multivariate model as well as the analysis of the individual- and community-level factors. 

We look forward to receiving your revised manuscript.

Kind regards,

Danielle Poole

Academic Editor

PLOS ONE

Journal Requirements:

2. We note you have included a table to which you do not refer in the text of your manuscript. Please ensure that you refer to Table 1 in your text; if accepted, production will need this reference to link the reader to the Table.

Reviewers' comments:

Reviewer's Responses to Questions

**Comments to the Author**

1. Is the manuscript technically sound, and do the data support the conclusions?

Reviewer #1: Yes

Reviewer #2: Partly

2. Has the statistical analysis been performed appropriately and rigorously? 

Reviewer #1: Yes

Reviewer #2: No

3. Have the authors made all data underlying the findings in their manuscript fully available?

Reviewer #1: Yes

Reviewer #2: No

4. Is the manuscript presented in an intelligible fashion and written in standard English?

Reviewer #1: Yes

Reviewer #2: Yes

5. Review Comments to the Author

Reviewer #1: The paper is well written and highlights the utilization of health facilities by youth, However, the authors seem to have laid a background on women, not youth specifically. The problem among this group need to be more pronounced. The methods have been well described. Were there limitations since this was secondary data? How was this overcome? The title could be changed to predictors instead of factors since that is what is entirely referred to in the whole paper. The results need to have n values. Line 260, correct ANV to ANC. Line 280, add full stop. The discussion is well done, it compares the findings well with other studies and also there is the author's attribution of the findings. The section on conclusion includes recommendations and it should be titled so.

Reviewer #2: I recommend the authors to use "multilevel logistic regression" analysis since there are community and individual factors rather than using "logistic regression". See the attached files for the detail.

6. PLOS authors have the option to publish the peer review history of their article (what does this mean?). If published, this will include your full peer review and any attached files.

Reviewer #1: **Yes: **Joyce Jebet Cheptum

Reviewer #2: **Yes: **Maru Mekie

---

## [Author Response · Author response to Decision Letter 0]

2 Jul 2021

We explored the use of multilevel modeling but the small sample of youth in each cluster could not allow the use of cluster as the second level of analysis.

This is because the Uganda Demographic and Health Surveys sampling is random by cluster and household levels. These would be the perfect levels for the multilevel analysis but both have few cases.

I have referred to table 1 in the text in line 141 in the clean copy of the manuscript

---

## [Decision Letter · Decision Letter 1]

3 Mar 2022

PONE-D-20-31194R1PREDICTORS OF HEALTH FACILITY DELIVERY AMONG UNMARRIED AND MARRIED YOUTH IN UGANDAPLOS ONE

Dear Dr. Agaba,

Thank you for submitting your manuscript to PLOS ONE. After careful consideration, we feel that it has merit but does not fully meet PLOS ONE’s publication criteria as it currently stands. Therefore, we invite you to submit a revised version of the manuscript that addresses the points raised during the review process.

 Reviewer 2 is not yet satisfied with the improvements and has detailed suggestions mostly for missing elements. In this respect, PLOS ONE endorses the use of the STROBE checklist (http://www.strobe-statement.org) to ensure completeness of reporting. Check also PLOS ONE guidelines for statistical reporting. Note that in your case you are making use of secondary data from standard sources (DHS). You do not need to delve in detail on the aspects connected the design, that you did not implement. You can point to the relevant sources, however.

We look forward to receiving your revised manuscript.

Kind regards,

José Antonio Ortega, Ph.D.

Academic Editor

PLOS ONE

Journal Requirements:

Reviewers' comments:

Reviewer's Responses to Questions

**Comments to the Author**

1. If the authors have adequately addressed your comments raised in a previous round of review and you feel that this manuscript is now acceptable for publication, you may indicate that here to bypass the “Comments to the Author” section, enter your conflict of interest statement in the “Confidential to Editor” section, and submit your "Accept" recommendation.

Reviewer #1: (No Response)

Reviewer #2: (No Response)

2. Is the manuscript technically sound, and do the data support the conclusions?

Reviewer #1: Yes

Reviewer #2: Yes

3. Has the statistical analysis been performed appropriately and rigorously? 

Reviewer #1: Yes

Reviewer #2: No

4. Have the authors made all data underlying the findings in their manuscript fully available?

Reviewer #1: Yes

Reviewer #2: Yes

5. Is the manuscript presented in an intelligible fashion and written in standard English?

Reviewer #1: Yes

Reviewer #2: No

6. Review Comments to the Author

Reviewer #1: The paper has been well revised and it captures the earlier suggested corrections. The paper can be accepted for publication, however the authors need to check grammatical and editorial errors prior.

Reviewer #2: REVIEW PLOS ONE

TITLE: FACTORS ASSOCIATED WITH HEALTH FACILITY DELIVERY AMONG UNMARRIED AND MARRIED YOUTH IN UGANDA

ARTICLE ID: PONE-D-20-31194R1

General comment

The authors tried to address all the issues raised. Hence, I would like to thank the authors for taking comments to improve the manuscript.

Issues to be cleared

Abstract

1. It would be good if the authors consider revising the result presentation to make it sound (It is good to compare findings among married and unmarried youths. But the presentation is not attractive and there is redundancy (“Versus” repeatedly mentioned).

Please try to present the result in abstract in reader friendly manner.

Main document

Methods

2. The authors tried to address the previous comments raised in the method section. However, the are other issues that shall be cleared. What study design have you used? The authors reported nothing regarding the study design used in the manuscript except describing the sampling technique used as well as the type of survey in the DHS data.

3. You have tried to compare the level of facility delivery among married and unmarried youths and the associated factors. Is it a comparative study? If so, is the analysis method appropriate?

4. If it is not comparative study there may not be a need to put the title as “Married and Unmarried youth”. Please clearly address these issues.

Results

5. Pages 9-10 in narration of the result as well as pages 10-12 in Table 2; please correct P = 0.000 to “P< 0.001”,

6. You are comparing facility delivery among married and unmarried youth but nothing has been stated about the proportion of married and unmarried youth in each of the variables in table to aside from putting “p” from cross tabulation. Please put the cross-tabulation findings by inserting additional columns in Table 2 “Married youth N(%) “ and Unmarried youth N(%) for each variable”

7. On pages 16, line 329, Table 3: “Multivariate analysis of factors….” Shall be corrected as “Multivariable analysis of factors…since the two terms.

8. Please consider to put the proportion of married and unmarried youth in each of the variables entered in the multivariable model in Table 3 (using cross-tabulation) so that it will be simple for readers to observe consistency as well as to compare variation in the proportion among married and unmarried youths (Example parity one among unmarried & Married, Parity two among married & unmarried …secondary education among married and unmarried etc.) It is a must to have an additional two columns in Table 3 for proportions of unmarried and married youth in each of the variable.

9. On pages 16-17, Table 3, you have put keys for

RC: Reference Category

CI: Confidence Interval

However, both are not listed as acronyms in Table 3 . Better to remove

Put 1: as reference category below Table 3.

Discussion

10. Page 19, lines 378-380 you explained “Unmarried youth were observed to use health facilities more than the married youth in Uganda between 2006 and 2016. This could be due to the confounding effect of higher parity as most married youth were pregnant for the second time or more.

However, nothing has been stated regarding the proportion of “parity among married and unmarried youth” either in the descriptive part of the result section or another area which makes difficult to confirm whether proportion of parity was higher among unmarried youth compared to the married one.

Conclusion and recommendation

Well written and all the comments are addressed.

7. PLOS authors have the option to publish the peer review history of their article (what does this mean?). If published, this will include your full peer review and any attached files.

Reviewer #1: **Yes: **Joyce Jebet

Reviewer #2: **Yes: **Maru Mekie

---

## [Author Response · Author response to Decision Letter 1]

23 Mar 2022

REVIEW PLOS ONE

TITLE: FACTORS ASSOCIATED WITH HEALTH FACILITY DELIVERY AMONG UNMARRIED AND MARRIED YOUTH IN UGANDA

ARTICLE ID: PONE-D-20-31194R1

We appreciate reviewer one for the positive comments. We have tried to improve the manuscript and the grammatical errors.

General comment 

The authors tried to address all the issues raised. Hence, I would like to thank the authors for taking comments to improve the manuscript. 

Issues to be cleared 

Abstract 

1. It would be good if the authors consider revising the result presentation to make it sound (It is good to compare findings among married and unmarried youths. But the presentation is not attractive and there is redundancy (“Versus” repeatedly mentioned).

Please try to present the result in abstract in reader friendly manner. 

Response: We have tried to edit the abstract to make it sound and attractive

Main document 

Methods

2. The authors tried to address the previous comments raised in the method section. However, the are other issues that shall be cleared. What study design have you used? The authors reported nothing regarding the study design used in the manuscript except describing the sampling technique used as well as the type of survey in the DHS data. 

Response: UDHS 2006, 2011 & 2016 are cross-sectional surveys so cross-sectional design was used

3. You have tried to compare the level of facility delivery among married and unmarried youths and the associated factors. Is it a comparative study? If so, is the analysis method appropriate?

Yes, this is a comparative study 

The analysis method is appropriate because it shows the similarities and differences in the use of health facilities at childbirth among unmarried and married youth

4. If it is not comparative study there may not be a need to put the title as “Married and Unmarried youth”. Please clearly address these issues. 

Response: This is a comparative study

Results

5. Pages 9-10 in narration of the result as well as pages 10-12 in Table 2; please correct P = 0.000 to “P< 0.001”, 

Response: This has been edited on page in line 197-204

6. You are comparing facility delivery among married and unmarried youth but nothing has been stated about the proportion of married and unmarried youth in each of the variables in table to aside from putting “p” from cross tabulation. Please put the cross-tabulation findings by inserting additional columns in Table 2 “Married youth N(%) “ and Unmarried youth N(%) for each variable”

Response: The first four columns in the previous table showed the descriptive distribution of unmarried and married youth but may be it was not clear. We have edited it as you suggested; page 10-11

7. On pages 16, line 329, Table 3: “Multivariate analysis of factors….” Shall be corrected as “Multivariable analysis of factors…since the two terms. 

Response: This has been changed to multivariable

8. Please consider to put the proportion of married and unmarried youth in each of the variables entered in the multivariable model in Table 3 (using cross-tabulation) so that it will be simple for readers to observe consistency as well as to compare variation in the proportion among married and unmarried youths (Example parity one among unmarried & Married, Parity two among married & unmarried …secondary education among married and unmarried etc.) It is a must to have an additional two columns in Table 3 for proportions of unmarried and married youth in each of the variable.

Response: I have included the proportions of married and unmarried youth who have had a childbirth from the health facility; page 13-15

9. On pages 16-17, Table 3, you have put keys for

RC: Reference Category

CI: Confidence Interval 

However, both are not listed as acronyms in Table 3 . Better to remove 

Put 1: as reference category below Table 3. 

Response: RC & CI have been deleted and 1 used as reference category

Discussion

10. Page 19, lines 378-380 you explained “Unmarried youth were observed to use health facilities more than the married youth in Uganda between 2006 and 2016. This could be due to the confounding effect of higher parity as most married youth were pregnant for the second time or more. 

However, nothing has been stated regarding the proportion of “parity among married and unmarried youth” either in the descriptive part of the result section or another area which makes difficult to confirm whether proportion of parity was higher among unmarried youth compared to the married one. 

Response: In the descriptive section, we showed the distribution of unmarried and married youth by parity; most unmarried youth (85.6%) were of parity one while most married youth (60.6%) were of parity two, line 178-181

Conclusion and recommendation 

Well written and all the comments are addressed.

---

## [Editor Report · Decision Letter 2]

25 Mar 2022

PREDICTORS OF HEALTH FACILITY CHILDBIRTH AMONG UNMARRIED AND MARRIED YOUTH IN UGANDA

PONE-D-20-31194R2

Dear Dr. Agaba,

We’re pleased to inform you that your manuscript has been judged scientifically suitable for publication and will be formally accepted for publication once it meets all outstanding technical requirements.

Kind regards,

José Antonio Ortega, Ph.D.

Academic Editor

PLOS ONE

Additional Editor Comments (optional):

It is felt that the issues raised were addressed and there is no need to send back to the reviewers. Congratulations!
---

## [Editor Report · Acceptance letter]

30 Mar 2022

PONE-D-20-31194R2 

Predictors of health facility childbirth among unmarried and married youth in Uganda 

Dear Dr. Agaba:

I'm pleased to inform you that your manuscript has been deemed suitable for publication in PLOS ONE. Congratulations! Your manuscript is now with our production department. 

Kind regards, 

on behalf of

Dr. José Antonio Ortega 

Academic Editor

PLOS ONE